# The Effect of Vanadium on Modified Z-Phase Characteristics in Austenitic Steels

**Vlastimil Vodárek** [1,*]**, Jan Holešinský** [2]**, Zdeněk Kuboň** [3]**, Renáta Palupčíková** [1]**, Kryštof Hradečný** [1]**, Petra Váňová** [1] **and Josef Hlinka** [1]

1 Department of Materials Engineering and Recycling, Faculty of Materials Science and Technology, VSB-Technical University of Ostrava, Poruba, 708 30 Ostrava, Czech Republic
2 ČEZ, a.s., Duhová 2/1444, 140 53 Prague, Czech Republic
3 MATERIAL AND METALLURGICAL RESEARCH, Ltd., Pohraniční 31, Vítkovice, 706 02 Ostrava, Czech Republic
* Correspondence: vlastimil.vodarek@vsb.cz

**Abstract:** In austenitic steels, the tetragonal Z-phase (NbCrN) has frequently been credited with beneficial strengthening effects during dislocation creep. In the modified Z-phase, niobium is partially substituted by vanadium. The basic objective of this contribution is a detailed characterization of the modified Z-phase in vanadium bearing austenitic AISI 316LN+Nb+V and HR3C steels. Experimental activities were focused on crystallography, thermodynamic and dimensional stability, kinetics of precipitation (TTP diagram) and solvus temperature of the modified Z-phase in the steels examined. Thermodynamic modelling was used for prediction of stable minor phases and solvus temperature of the modified Z-phase. Kinetics of precipitation of the (Nb,V)CrN phase in the AISI 316LN+Nb+V steel was experimentally investigated in the temperature interval of 550–1250 °C. The kinetics of precipitation of the modified Z-phase in austenitic matrix was fast. Results of diffraction studies on particles of the modified Z-phase confirmed the existence of the tetragonal unit cell already after short-term annealing. The solvus temperature of the modified Z-phase in austenitic steels was determined to be lower than that for the NbCrN phase. The decrease in the solvus temperature is dependent on the vanadium content in austenitic steels. Both thermodynamic calculations and experimental results proved that the thermodynamical stability of the modified Z-phase in austenite was high. More data are needed for evaluation of long-term dimensional stability of the (Nb,V)CrN phase in austenitic steels at temperatures for their engineering applications.

**Keywords:** AISI 316LN+Nb+V; Super 304H; modified Z-phase; crystal structure; solvus temperature; thermodynamic stability; dimensional stability

## 1. Introduction

In last decades, service conditions of components for advanced power plants have increased, imposing more severe requirements on creep properties and corrosion resistance of high temperature materials [1–3]. Much effort has been devoted to the development of new heat resistant steels, both martensitic and austenitic, and to better understand their microstructure–property relationships. Small changes in the chemical composition of heat resistant steels can lead to substantial changes in their creep resistance [1]. Creep properties of heat resistant steels in the field of dislocation creep are controlled by the evolution of dislocation substructure, and the formation and growth of sub-grains [4]. Evolution of dislocation structure is strongly affected by the formation, coarsening and dissolution of minor phases [3]. Generally, the matrix solid solution and the precipitation hardening associated with a high density of fine, both thermodynamically and dimensionally stable, precipitates provide resistance to dislocation creep. Increasing the extent of precipitation strengthening is one of the most successful methods for improving creep resistance of high

temperature materials [1]. Although many minor phases are now well documented, there are still contradictions and missing thermodynamic data about some minor phases [5,6].

Advanced austenitic heat resistant steels are often alloyed by niobium and nitrogen, which allow it to precipitate a complex nitride called Z-phase (NbCrN) [7–18]. Fine particles of this minor phase in austenitic heat resistant steels have frequently been credited with beneficial strengthening during creep [7,11]. Conditions for the Z-phase formation are not clear. Robinson and Jack [19] reported the formation of Z-phase from solid solution, while Knowles [20] described its formation from MX precipitates. The results must be examined carefully as the steel compositions were different. The solvus temperature of the NbCrN phase in austenitic stainless steels was reported to be between 1250 and 1350 °C, depending on the steel composition [6]. Z-phase particles in the AISI 316LN+Nb steel nucleated from solid solution and formed short rods elongated in the $[001]_Z$ direction. This is in accordance with the minimum directional mismatch, but some particles exhibited roughly comparable growth in the $[001]_Z$ direction and in the $<100>_Z$ directions lying in the base plane [21]. In advanced austenitic steels, Z-phase often nucleates on MX (NbN) particles and forms thin plates [22,23].

The structure of Z-phase was studied by Ettmayer [24] and by Jack and Jack [25]. Ettmayer [24] studied the synthesized complex nitrides NbCrN and reported a tetragonal unit cell with parameters a = 0.4283 nm and c = 0.7361 nm, space group F4bm. Jack and Jack [25] studied Z-phase in an austenitic steel and interpreted all the X-ray data by a tetragonal cell of dimensions a = 0.3037 nm, c = 0.7391 nm. The relationship between this unit cell and that of the Ettmayer's cell in the basal plane is $\sqrt{2}a_{JACK} = a_{ETTM.}$. For the unit cell proposed by Jack and Jack [25] systematic absences occur for hk0 when h + k = 2n + 1. Trial and error methods showed that Z-phase is a complex nitride with ideal composition $Nb_2Cr_2N_2$. The metal atom arrangement is characterised by double layers of similar atoms alternating along the c axis to provide an AABBAABBAA sequence.

In martensitic (9–12) %Cr steels, niobium in the Z-phase has been found to be partially substituted by vanadium and this results in a reduction of the tetragonal unit cell of this modified Z-phase ((Nb,V)CrN): a = 0.286 nm and c = 0.739 nm) [26]. In the temperature range of 600–650 °C, the Z-phase is the most stable nitride in (9–12) %Cr steels. The formation of the modified Z-phase is slow and is associated with the decomposition of fine nitrides of MX ((Nb,V)N), which significantly contributes to precipitation hardening [26–28]. The modified Z-phase forms large particles and is generally regarded as a harmful minor phase in these steel grades [3].

Extensive studies revealed that the FCC precursor with a lattice parameter a = 0.405 nm can coexist with the tetragonal unit cell of the Z-phase [29,30]. It was reported that during ageing of 12CrMoVNbN steels at 600 °C, the cubic unit cell was predominant for times of exposure of the order of $10^4$ h, while the tetragonal unit cell was more frequent upon prolonged ageing [31]. The FCC unit cell is regarded as an intermediate metastable crystal structure and is expected to be gradually replaced by the tetragonal unit cell of the Z-phase [30,31]. The relationship between the tetragonal and cubic unit cells in hybrid (FCC/Z-phase) particles is shown in Figure 1.

Much effort has been devoted to investigations on nucleation, growth and chemical composition of this phase in martensitic (9–12) %Cr steels [32–37]. Chromium content plays a critical role in the driving force for the Z-phase precipitation. Detailed investigations proved that the dominant mechanism of the modified Z-phase formation is related to in situ transformation of (Nb,V)N particles into the FCC precursor by diffusion of chromium from the surroundings [30,32,35,36]. After the development of chromium rich regions at rims of the host MX particle, these regions consume vanadium, niobium and nitrogen from the dissolving MX particle. Experimental results are consistent with this mechanism of gradual conversion of MX precipitates via the FCC precursor rather than independent nucleation of Z-phase followed by dissolution of MX particles [30,32]. Final transformation of the FCC precursor to the tetragonal Z-phase is associated with ordering of solute atoms to produce the double layer structure which is typical for the tetragonal Z-phase [30,38].

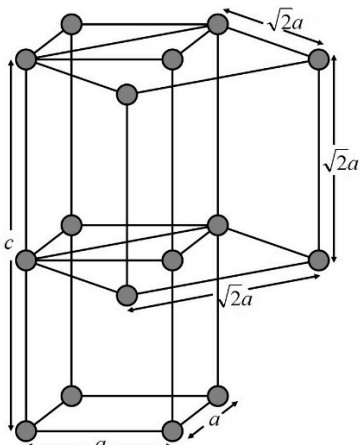

**Figure 1.** The relationship between the tetragonal unit cell of the Z-phase and the FCC unit cell [29].

Vanadium is usually not reported in the nominal chemical composition of advanced austenitic steels. However, results of detailed chemical analyses prove that commercial heats of these steels often contain small amounts of vanadium. The presence of vanadium creates prerequisites for the partial substitution of niobium with vanadium in the particles of the Z-phase. A lack of information exists about precipitation behaviour of the modified Z-phase in austenitic steels containing both niobium and vanadium additions. Particles with the chemical composition corresponding to (Nb,V)CrN phase were observed after isothermal annealing at 750 °C/1170 h in an austenitic 18Cr-12Ni-VNbN steel in [39]. These particles were not reported as Z-phase because their crystal structure corresponded to FCC NaCl structure. Karlsson et al. [39] speculated that the particles with the FCC crystal structure could be a precursor of Z-phase.

This paper deals with basic characteristics of the modified Z-phase in the AISI 316LN+Nb+V steel and in the HR3C steel.

## 2. Materials and Methods

Chemical composition of a laboratory heated AISI 316LN+Nb+V steel is shown in Table 1. Preliminary heat treatment of forged rods with dimensions of ϕ12 × 125 mm consisted of solution annealing at 1300 °C for 0.5 h followed by quenching in water. Next, annealing of test samples was carried out at the temperature interval of 550 to 1250 °C with a step of 100 °C for 1, 20 and 50 h, followed by water quenching. Detailed microstructural characterization of annealed samples was performed. The results of minor phase identification were used for the construction of the TTP diagram of the modified Z-phase precipitation in the austenitic matrix.

**Table 1.** Chemical composition of the AISI 316LN+Nb+V steel, wt.%.

| C | S | Mn | Si | P | Cr | Ni | Mo | V | Nb | N |
|------|-------|------|------|-------|-------|-------|------|------|------|------|
| 0.03 | 0.007 | 1.44 | 0.59 | 0.017 | 18.10 | 12.50 | 2.44 | 0.25 | 0.11 | 0.16 |

Chemical composition of an industrial heat of the HR3C steel is shown in Table 2. As evident, a small amount of vanadium is present. Solution annealing of the tube ϕ42.8 × 6 mm was performed at 1250 °C for 2 min. The effect of thermal exposure on the Z-phase characteristics was studied on the sample that was annealed at 650 °C for 8600 h.

**Table 2.** Chemical composition of the HR3C steel, wt.%.

| C | S | Mn | Si | P | Cu | Cr | Ni | Mo | Nb | V | N |
|------|-------|------|------|-------|------|-------|-------|------|------|-------|-------|
| 0.05 | 0.003 | 1.18 | 0.37 | 0.013 | 0.06 | 25.27 | 20.17 | 0.17 | 0.45 | 0.032 | 0.256 |

Minor phase identification in the studied austenitic steels was performed using transmission electron microscopy (TEM). Both carbon extraction replicas and thin foils were used. Electron microscopy studies were performed on a JEM 2100 microscope (JEOL, Tokyo, Japan), equipped with INCA EDX system. The identification of minor phases was performed using a combination of selected area electron diffraction (SAED) and energy-dispersive X-ray microanalysis (EDX). The holder of the JEM 2100 microscope enabled double tilt in the range of angles x = ±30°, y = ±30°. SAED patterns at controlled tilt of the holder were obtained. Diffraction experiments revealed that Kikuchi patterns from (Nb,V)CrN particles in low index orientations are very similar to Kikuchi patterns of FCC crystals. This fact allowed for the use of the Kikuchi patterns of an FCC crystal for controlled navigation from one zone axis to another. Controlled tilting of particles into required orientations was very important in order to perform a reliable identification of the unit cell of particles investigated. The analysis was focused on attaining diffraction patterns exhibiting some reflections belonging solely to the tetragonal Z-phase, i.e., extra spots. Diffraction studies were accompanied by semiquantitative EDX microanalyses. Interstitial elements were not included into quantification of EDX spectra and results were normalized to 100%. The criterion applied for identification of the modified Z-phase was: 40 at.% < [Fe] + [Cr] < 70 at.% [5,40]. For preparation of carbon extraction replicas, etched metallographic samples were coated with a carbon layer with a thickness of 12 nm. Electrolytic etching in a 5% HCl solution in alcohol was applied to release the carbon film from the substrate. The carbon film with minor phase particles was stripped to the water surface, and after a rinse in water, pieces of carbon film were fished out on the backing grid. The thin foils were prepared by twin jet electropolishing in a solution of 5%HClO$_4$ in glacial acetic acid at room temperature and a voltage of 80 V.

For reliable interpretation of experimental spot diffraction patterns, kinematical SAED patterns for both the tetragonal unit cell and the FCC unit cell were simulated using a computer programme. In calculations of kinematical intensities of Z-phase reflections, niobium atoms were replaced by vanadium, i.e., calculations were performed for VCrN phase. It was assumed that the positions of individual atoms in the unit cell were the same as those in the NbCrN phase [25].

Thermocalc software (Thermo-calc software, Stockholm, Sweden) and the TCFE 8 database were used for prediction of equilibrium minor phases in the steels investigated. Attention was paid to the effect of vanadium in the modified Z-phase on the solvus temperature of this phase in austenite.

## 3. Results

### 3.1. Simulation of Spot Diffraction Patterns

The relationship among coordinate axes of the modified Z phase and the FCC precursor, see Figure 1, can be expressed as follows:

$$[100]_Z = \frac{1}{2}\left[1\,\overline{1}\,0\right]_{FCC}; \quad [010]_Z = \frac{1}{2}[110]_{FCC}; \quad [001]_Z = \frac{c_Z}{a_{FCC}}[001]_{FCC} = 1.83[001]_{FCC} \quad (1)$$

These equations determine elements of transformation matrices which can be applied for transformation of Miller indexes of any vector $\vec{r}$ in real space:

$$\left[FCC, \vec{r}\right] = (FCC\mathbf{T}Z)\left[Z, \vec{r}\right] \quad \left[FCC, \vec{r}\right] = \begin{pmatrix} 0.5 & 0.5 & 0 \\ -0.5 & 0.5 & 0 \\ 0 & 0 & 1.83 \end{pmatrix}\left[Z, \vec{r}\right] \quad (2)$$

$$\left[Z, \vec{r}\right] = (Z\mathbf{T}FCC)\left[FCC, \vec{r}\right] \quad \left[Z, \vec{r}\right] = \begin{pmatrix} 1 & -1 & 0 \\ 1 & 1 & 0 \\ 0 & 0 & 0.546 \end{pmatrix}\left[FCC, \vec{r}\right] \quad (3)$$

For transformation of Miller indexes of vector $\vec{g}$ in reciprocal space, the following transformation matrices can be used:

$$\left[Z,\vec{g}\right]^* = \begin{pmatrix} 0.5 & -0.5 & 0 \\ 0.5 & 0.5 & 0 \\ 0 & 0 & 1.83 \end{pmatrix} \left[FCC,\vec{g}\right]^* \quad \left[FCC,\vec{g}\right]^* = \begin{pmatrix} 1 & 1 & 0 \\ -1 & 1 & 0 \\ 0 & 0 & 0.546 \end{pmatrix} \left[Z,\vec{g}\right]^* \quad (4)$$

Transformation matrices were applied for calculations of all possible variants of diffraction patterns of the tetragonal unit cell of the Z-phase corresponding to the zone axes of the FCC precursor with low Miller indexes. The results of calculations for some low index zone axes are shown in Table 3. In some zone axes, the differences between the corresponding FCC and Z-phase diffraction patterns are so small that these zone axes should not be used to reliably distinguish between both phases by electron diffraction techniques. It means that in some zone axes all spots of the FCC phase are very close to Z-phase spots. Only in a limited number of Z-phase orientations are there extra spots in SAED patterns, i.e., spots belonging solely to the Z-phase. Such diffraction patterns make it possible to differentiate reliably between the tetragonal modified Z-phase and the FCC phase. As a rule, if the occurrence of some spots in diffraction patterns cannot be explained using the FCC unit cell, then the FCC unit cell is not the correct choice for interpretation of diffraction results. Figure 2a,b show the simulated diffraction patterns for zone axes [111]$_{FCC}$ and [041]$_Z$, respectively. In the [041]$_Z$ zone axis there are extra spots which make it possible to uniquely distinguish between the tetragonal Z-phase and the FCC precursor.

**Table 3.** Transformation of the FCC low index zone axes of the precursor into the tetragonal unit cell of the modified Z-phase.

| <uvw>$_{FCC}$ | [uvw]$_{FCC}$ | [uvw]$_Z$ | Extra Spots * |
|---|---|---|---|
| <100> | [100] | [110] | Yes |
|  | [001] | [001] | No |
| <110> | [110] | [010] | Yes |
|  | [101] | [221] | No |
| <111> | [111] | [041] | Yes |

* the occurrence of extra spots in diffraction patterns of the tetragonal Z-phase.

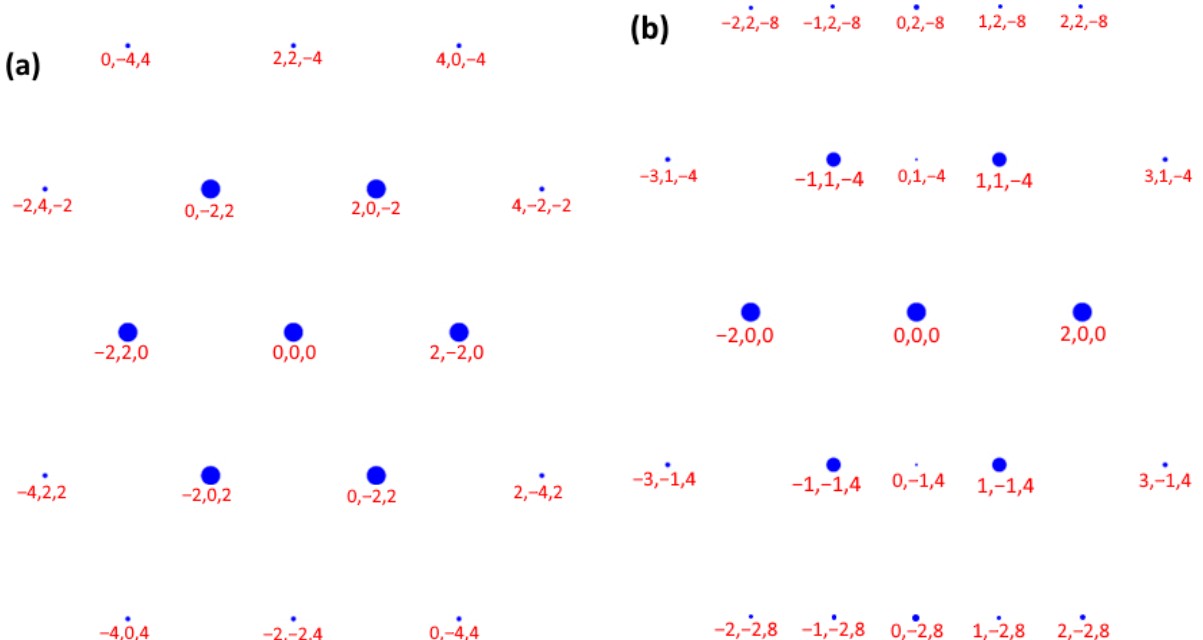

**Figure 2.** Simulated spot diffraction patterns: (**a**) zone axis [111]$_{FCC}$, (**b**) zone axis [041]$_Z$.

It is worth noting that in many cases simulated intensities of the extra spots are very weak, and therefore, it might be difficult to observe some of these spots in experimental diffraction patterns. Furthermore, simulated intensities of potentially overlapping spots of the tetragonal Z-phase and the FCC phase are not implicitly stronger than intensities of extra spots. The results of SAED simulations suggest that reliably distinguishing between the FCC and tetragonal unit cells under consideration requires a series of diffraction patterns obtained on each (Nb,V)CrN particle investigated.

### 3.2. Investigations on the Modified Z-Phase in the AISI 316LN+Nb+V Steel

TEM analysis of annealed samples revealed only two minor phases: MX and niobium, vanadium and chromium rich particles of the modified Z-phase. In samples annealed at 1250 °C neither MX nor modified Z-phase were present. It proved that preliminary heat treatment was carried out above the solvus temperature of these minor phases. Particles of MX and the modified Z-phase could not be distinguished on their morphology alone. Particles of MX phase were identified in all samples annealed in the temperature interval of 650–1150 °C. Figure 3 shows an MX particle in the sample after annealing at 1150 °C for 50 h. The average chemical composition of the MX phase at temperatures of 750, 950 and 1150 °C is shown in Table 4. Except for vanadium and niobium, particles of this phase also contained some chromium. The average content of chromium in the MX phase decreased with growing annealing temperature. Some MX particles exhibited needle-like morphology. The average size of MX particles decreased with decreasing temperature and time of annealing. MX particles decorated austenite grain boundaries and were also present inside austenite grains. Figure 4 shows intragranular precipitation of MX particles and the corresponding ring diffraction pattern in the sample annealed at 850 °C for 1 h. The lattice parameter of the FCC unit cell of the MX phase was determined as a = 4.19 nm. In all specimens investigated, MX particles constituted most precipitates.

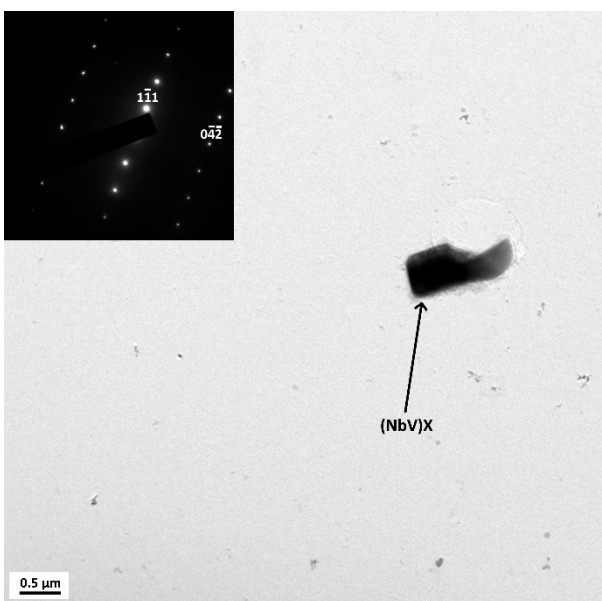

**Figure 3.** MX particle in the sample after annealing 1150 °C/50 h, bright field (BF) image; insert: zone axis [321]$_{MX}$, carbon extraction replica.

**Table 4.** Results of semiquantitative EDX analyses on MX particles; annealing time: 20 h, at.%.

| Temperature [°C] | V | Cr | Nb |
|---|---|---|---|
| 750 | 27.1 ± 1.8 | 37.3 ± 5.4 | 35.6 ± 6.3 |
| 950 | 44.5 ± 5.3 | 19.3 ± 3.9 | 36.2 ± 4.4 |
| 1150 | 36.2 ± 1.3 | 16.7 ± 1.5 | 49.1 ± 1.2 |

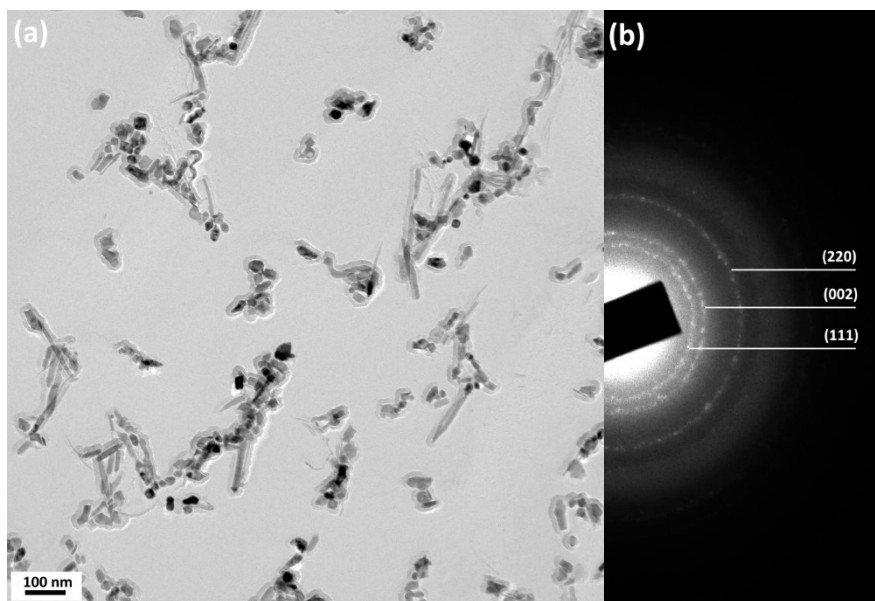

**Figure 4.** Intragranular precipitation after annealing 850 °C/1 h. (**a**) BF image, carbon extraction replica, (**b**) ring diffraction pattern of the MX phase ((Nb,V)N).

Particles of the modified Z-phase were identified in the temperature interval of 750–1050 °C. In this temperature range, particles of the modified Z-phase were present in all samples annealed for 20 and 50 h. At temperatures of 850 °C and 950 °C, particles of this minor phase were detected after holding time for one hour. Particles of the modified Z-phase in some cases nucleated on MX particles (Figure 5a). Diffraction analysis revealed that the crystal structure of this particle corresponds to the tetragonal Z-phase. Figure 5c–e show the SAED spot diffraction pattern and simulated diffraction patterns for $[010]_Z$ and $[110]_{FCC}$ of the FCC precursor, respectively. Due to extra spots in the diffraction pattern belonging to the tetragonal Z-phase, it is easy to discriminate between the tetragonal and FCC unit cells. Extra spots (in circles) in the SAED diffraction pattern prove that the crystal structure of the particle is tetragonal (Figure 5c,d). Streaking of reflections in the [001]* direction is probably related to defects on (001) atomic planes due to imperfect ordering of solute atoms in the double layer structure of the Z-phase. This effect was not observed in diffraction patterns on particles which precipitated at higher temperatures or longer holding times. Figure 5b shows the dark field (DF) image in the reflection $10\bar{2}_Z$.

Figure 6a shows intragranular precipitation in the sample annealed at 1050 °C for 50 h, which was the maximum temperature for the formation of the modified Z-phase. Spot diffraction pattern in Figure 6b matches the simulated diffractogram in Figure 6c which belongs to the tetragonal unit cell of the Z-phase. The corresponding simulated diffraction pattern of the FCC phase is shown in Figure 6d.

Particles of the modified Z-phase formed at 750 °C showed the needle-like morphology (several nanometres in width and (50–100) nanometres in length). The average particle size increased with the annealing temperature and time. After annealing at 1050 °C, particles of the modified Z-phase reached up to several hundreds of nanometres in length. More data are needed for evaluation of long-term dimensional stability of the (Nb,V)CrN phase in austenitic steels at temperatures for their engineering applications.

Figure 7 summarises results of investigations on the kinetics of the modified Z-phase formation in the temperature interval of 550–1250 °C. The nose of the C-curve shows that the fastest precipitation of the modified Z-phase can be expected at approximately 900 °C. The results of experimental studies indicate that the solvus temperature of the modified Z-phase in the AISI 316LN+Nb+V steel is approximately 1050 °C.

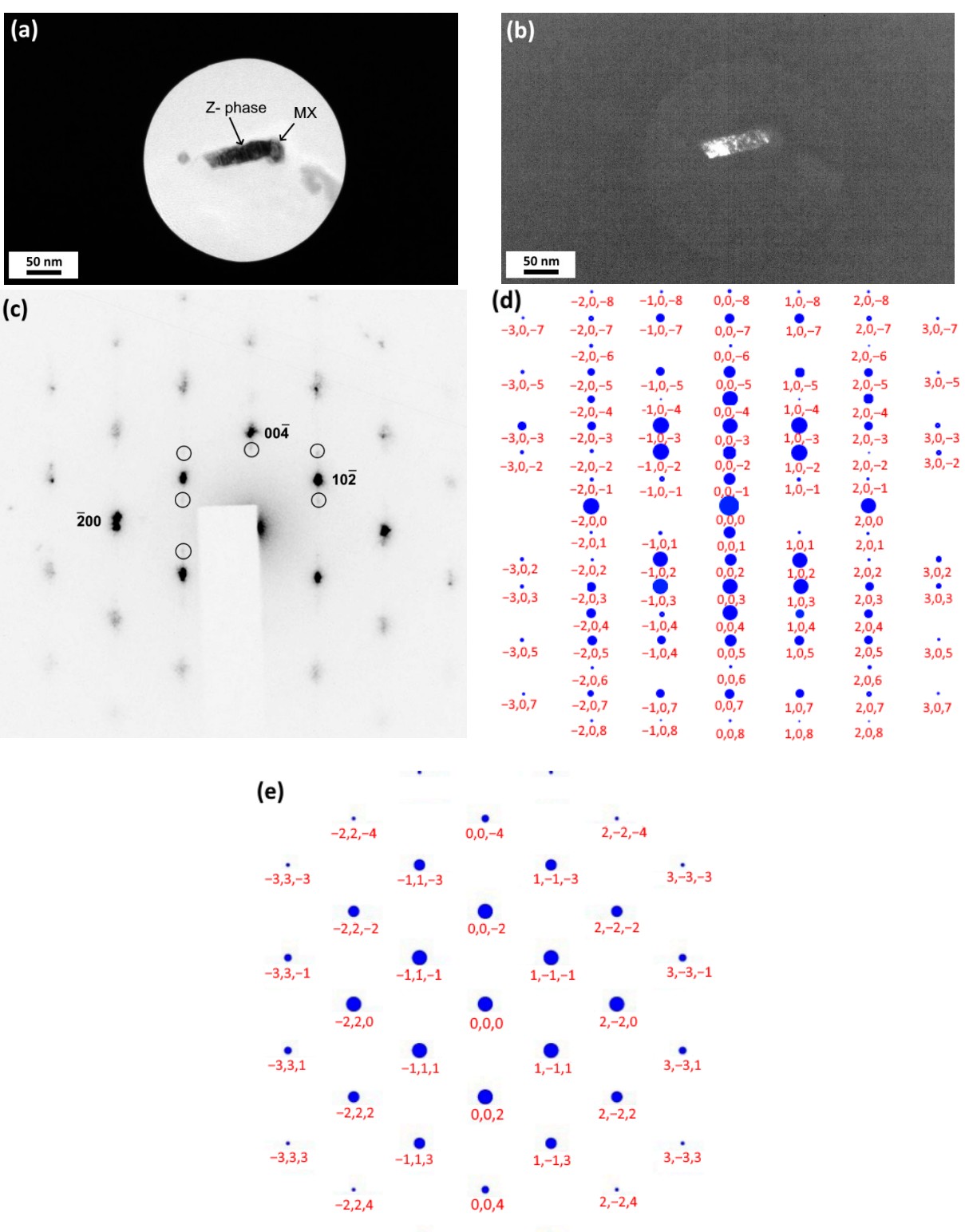

**Figure 5.** Diffraction analysis of the Z-phase particle nucleated on the MX particle. (**a**) BF image, (**b**) DF image in $10\bar{2}_Z$, (**c**) SAED pattern, (**d**) simulated spot diffraction pattern for zone axis $[010]_Z$, (**e**) simulated diffraction pattern for zone axis $[110]_{FCC}$.

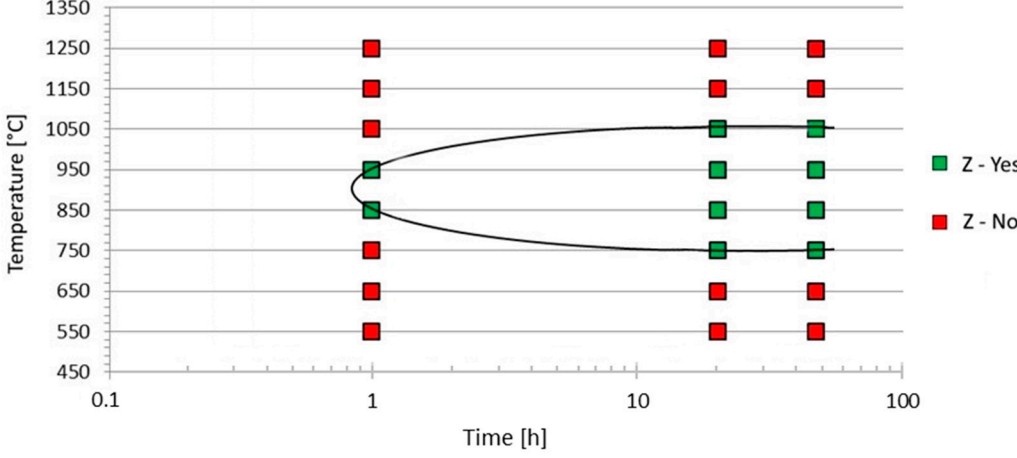

**Figure 6.** Analysis of the Z-phase particle in the sample after annealing 1050 °C/20 h. (**a**) Intragranular precipitation, BF image, carbon extraction replica, (**b**) spot diffraction pattern, (**c**) simulated diffraction pattern for the zone axis [021]$_Z$, (**d**) simulated diffraction pattern for the zone axis [112]$_{FCC}$.

**Figure 7.** TTP diagram of the modified Z-phase in the AISI 316LN+Nb+V steel.

Results of semiquantitative EDX analyses of the modified Z-phase in samples annealed at temperatures in the range of 750–1050 °C for 20 h are summarized in Table 5. As the annealing temperature increases, the chromium content in the modified Z-phase decreases. Furthermore, the Nb/V ratio increases with increasing annealing temperature.

**Table 5.** Results of semiquantitative EDX analyses on the modified Z-phase; annealing time: 20 h, at.%.

| Temperature [°C] | Cr | Fe | Mo | Nb | V |
|---|---|---|---|---|---|
| 750 | 45.9 ± 4.8 | 5.9 ± 1.4 | 3.6 ± 1.1 | 17.6 ± 3.8 | 27.0 ± 2.3 |
| 850 | 42.3 ± 1.8 | 4.0 ± 0.6 | 2.1 ± 0.6 | 17.5 ± 3.1 | 34.1 ± 2.1 |
| 950 | 37.3± 0.6 | 3.7 ± 0.4 | 2.6 ± 0.4 | 28.5 ± 1.1 | 28.0 ± 1.1 |
| 1050 | 35.7 ± 3.6 | 4.7 ± 0.3 | 2.7 ± 1.0 | 44.0 ± 4.4 | 12.8 ± 1.7 |

*3.3. Thermodynamic Modelling*

Simulation of the thermodynamic equilibrium in the AISI 316LN+Nb+V steel was carried out using the Thermocalc software and the TCFE 8 database. Figure 8 shows the dependence of the amount of equilibrium minor phases on temperature. The results predict that the modified Z-phase is stable up to the temperature of 1055 °C. The predicted solvus temperature of the modified Z-phase in the AISI 316LN+Nb+V steel is approximately 200 °C lower than that of the NbCrN phase in niobium- and nitrogen-bearing austenitic steels. The MX phase is predicted as a thermodynamically stable minor phase at temperatures above approximately 1050 °C.

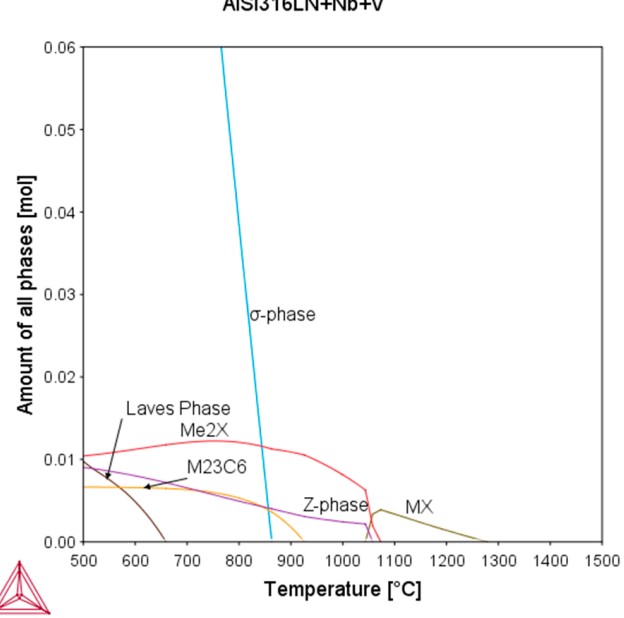

**Figure 8.** Thermocalc simulation of the amount of equilibrium phases on temperature for AISI 316LN+Nb+V steel.

Table 6 shows calculated equilibrium compositions of the modified Z-phase in the AISI 316LN+Nb+V steel at 750 and 1050 °C. The equilibrium vanadium and chromium contents in the modified Z-phase decrease with increasing temperature, while the niobium content increases with increasing temperature. The compositional change trends correspond to the results of the experimental study which are presented in Table 5.

**Table 6.** Calculated equilibrium composition of the modified Z-phase in the AISI 316LN+Nb+V steel, Thermocalc, TCFE 8 database, at.%.

| Temperature [°C] | Cr | Fe | Mo | Nb | V |
|---|---|---|---|---|---|
| 750 | 44.6 | 5.4 | 0.3 | 15.9 | 33.8 |
| 1050 | 43.2 | 6.8 | 2.4 | 30.0 | 17.6 |

*3.4. Investigations on the Modified Z-Phase in the HR3C Steel*

The effect of a small amount of vanadium in the commercial HR3C steel on precipitation reactions was studied after thermal exposure at 650 °C for 8600 h. Table 2 shows that the tube investigated contained 0.03 wt.% of vanadium.

TEM investigations revealed the following minor phases: σ-phase, $M_{23}C_6$, NbN and the modified Z-phase. Figure 9a,b documents modified Z-phase particles in the form of thin plates which were nucleated on NbN particles along dislocations. This mechanism of the formation of the Z-phase in a 25Cr-20Ni-Nb-N austenitic stainless steel was reported by Li et al. [23]. Gradual dissolution of NbN particles supplies the necessary niobium and nitrogen for growth of NbCrN particles. The supply of niobium and nitrogen over a short distance enabled relatively fast growth of modified Z-phase particles. As can be seen in Figure 9b, the typical length of Z-phase particles was 50–100 nm. Distribution of the modified Z-phase particles in austenitic grains was very heterogeneous, forming clusters of side-by-side thin plates along dislocations.

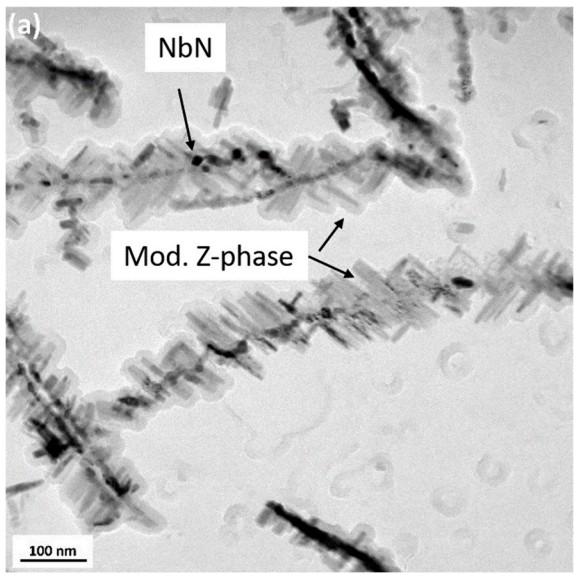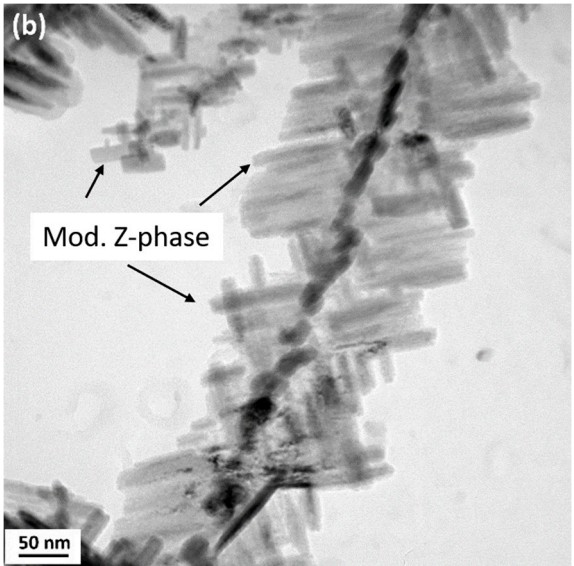

**Figure 9.** Precipitation of the modified Z-phase in the HR3C steel, thermal exposure: 650 °C/8600 h. (**a**) Nucleation of Z-phase on NbN particles (dark contrast) along dislocations, (**b**) detail of modified Z-phase thin plates arranged along dislocation lines, BF images, carbon extraction replica.

EDX investigations confirmed that due to a small amount of vanadium in the HR3C steel, particles of the Z-phase contained some vanadium. As evident from Table 7, the average content of vanadium in the modified Z-phase was approximately 3 wt.%. The vanadium content in the modified Z-phase in the HR3C steel was significantly lower than that in the AISI 316LN+Nb+V steel. This is related to differences in vanadium content between the two steels.

**Table 7.** Results of semiquantitative EDX analyses of the modified Z-phase in the HR3C steel (at.%).

| V | Cr | Fe | Nb | Mo |
|---|---|---|---|---|
| 3.2 ± 1.3 | 51.0 ± 1.2 | 2.9 ± 0.9 | 38.6 ± 4.3 | 4.3 ± 2.7 |

Results of the thermodynamic modelling of the solvus temperature of the modified Z-phase in the HR3C steel are shown in Figure 10. The solvus temperature is approximately 1200 °C. The MX (NbN) phase is predicted as a stable nitride above 1050 °C.

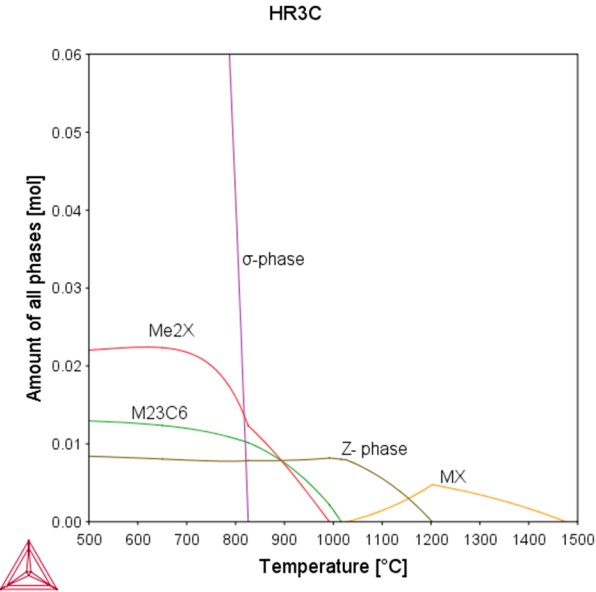

**Figure 10.** Thermocalc simulation of the amount of equilibrium phases on temperature for HR3C steel.

## 4. Discussion

The results of experimental studies on the AISI 316LN+Nb+V steel proved that the kinetics of the modified Z-phase ((Nb,V)CrN) precipitation in the temperature interval 750–1050 °C was fast. Nucleation of (Nb,V)CrN particles was observed both on MX ((Nb,V)N) particles and from solid solution. Both experimental results and Thermocalc calculations revealed that substitution of niobium by vanadium decreased with increasing temperature. Diffraction studies proved that unlike the modified Z-phase in (9–12) %Cr martensitic steels, the crystal structure of this complex nitride was already tetragonal after short annealing times. The constructed TTP diagram indicates that the solvus temperature of the (Nb,V)CrN phase in the steel investigated is approximately 1050 °C. This finding is in accordance with the prediction of the Thermocalc software. From the results of thermodynamic calculations, it follows that up to approximately 1050 °C, the thermodynamically stable nitride in the AISI 316LN+Nb+V steel is the modified Z-phase, and above the solvus temperature of this phase, the MX ((NbV)N) phase is predicted as a stable nitride.

The studied commercial heat of the HR3C steel contained approximately 0.03 wt.% of vanadium. After thermal exposure at 650 °C for 8600 h, the content of vanadium in particles of the modified Z-phase was approximately 3 wt.%. Precipitation of the (Nb,V)CrN particles was preceded by NbN particles, which preferentially nucleated on dislocations. The modified Z-phase particles formed on NbN precipitates and their growth was facilitated by transport of niobium and nitrogen from dissolving NbN particles. Particles of the (Nb,V)CrN phase formed thin plates. They precipitated side by side along dislocations (see Figure 9b). The typical length of Z-phase particles after thermal exposure 650 °C/8600 h was 50–100 nm. Thermocalc calculations predict that the solvus temperature of the modified Z-phase in the heat investigated is approximately 1200 °C. Above 1050 °C, the MX (NbN) phase is predicted as a stable nitride.

Distribution and size of Z-phase particles in CrNi(Mo) austenitic steels is significantly affected by the mechanism of nucleation of Z-phase particles and chemical composition of steels. A small addition (0.1 wt.%) of niobium to the AISI 316LN steel resulted in a significant reduction of the minimum creep rate and shortening of the tertiary creep stage [7]. The minimum creep rate reduction can be attributed to precipitation of fine particles of the secondary Z-phase (NbCrN). These particles nucleated from solid solution, mainly

on dislocations. The kinetics of the Z-phase formation was fast. Long-term dimensional stability of secondary Z-phase particles during creep/thermal ageing in the temperature interval of 600–650 °C was excellent. No NbN particles were found to precipitate during long-term ageing or creep in this temperature interval. The average size of secondary Z-phase particles in the AISI 316LN + 0.1%Nb steel after ageing 625 °C/102,600 h was determined as d = 11 ± 4 nm. Figure 11a shows pinning of dislocations by particles of the secondary Z-phase in the head of the creep sample ruptured after exposure 650 °C/55,500 h. Rectangular primary Z-phase particles (present in the matrix after solution annealing) and fine secondary Z-phase particles (formed during thermal exposure) in the sample after thermal exposure 650 °C/58,950 h are documented in Figure 11b. Distribution of these very fine secondary Z-phase particles in austenite is relatively homogeneous.

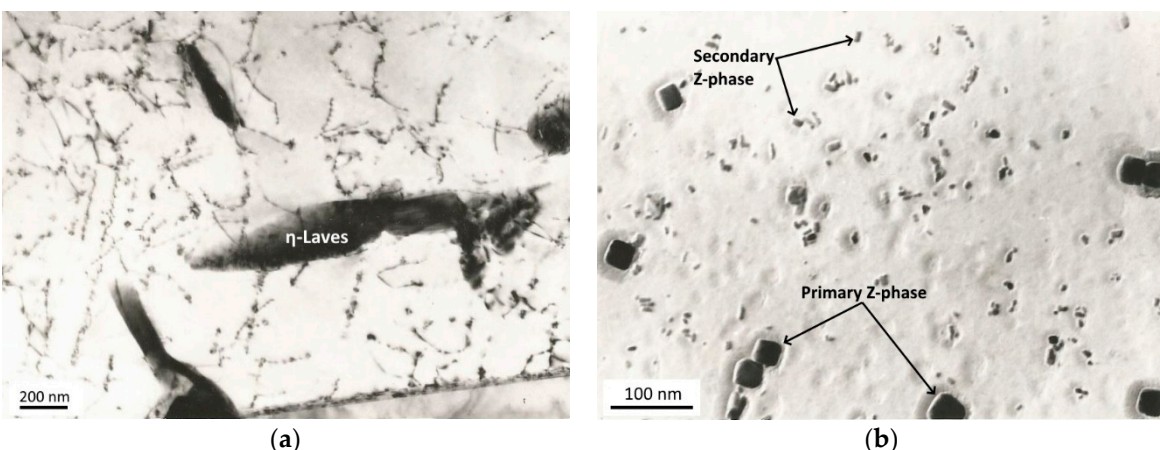

(a)                                             (b)

**Figure 11.** Precipitation of secondary Z-phase particles. (**a**) BF image, pinning of dislocations by fine secondary Z-phase particles, thermal exposure: 600 °C/55,500 h, thin foil, AISI 316LN + 0.3%Nb. (**b**) Distribution of primary and secondary Z-phase particles, 650 °C/58,950 h, carbon extraction replica, AISI 316LN + 0.1%Nb.

Figures 9 and 11 demonstrate that morphology and distribution of Z-phase particles in the HR3C and AISI 316LN+Nb steels are significantly different. The growth rate of (Nb,V)CrN particles in the HR3C steel during thermal exposure at 650 °C was significantly faster than that of NbCrN particles in the AISI 316LN+Nb steel.

Figure 12 shows results of the Thermocalc simulation of the amount of equilibrium phases on temperature in the AISI 316LN+Nb steel. The solvus temperature of the Z-phase is approximately 1250 °C. Microstructural characterization after solution annealing in the temperature interval of 1050–1250 °C confirmed the presence of primary Z-phase particles. The typical size of these particles was approximately 100 nm.

Experimental results demonstrate that fine intragranular Z-phase particles can have a positive effect on the minimum creep rate, but the final effect of niobium on long-term creep properties will also depend on its influence on the stability of other minor phases in the austenitic matrix [7]. The positive effect of niobium in the AISI 316LN steel on the creep resistance in the first and second stages of creep was gradually surpassed by its effect on acceleration of the σ-phase, $M_6X$ ($Cr_3Ni_2SiX$ type) and η-Laves ($Fe_2Mo$ type) formation. Coarse σ-phase particles at austenite grain boundaries promoted the formation of creep defects [7].

The presented results prove that vanadium in nitrogen- and niobium-bearing austenitic steels dissolves in the modified Z-phase ((Nb,V)CrN) and lowers its solvus temperature. This can help reduce the fraction of primary Z-phase particles in the austenitic matrix after solution annealing. More data are needed for evaluation of the long-term dimensional stability of (Nb,V)CrN particles in austenitic steels at temperatures for their engineering applications.

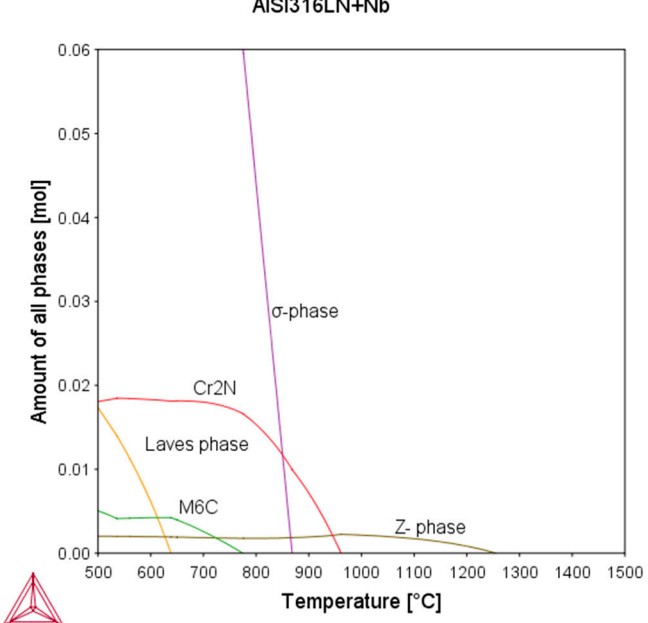

**Figure 12.** Thermocalc simulation of the amount of equilibrium phases on temperature, AISI 316LN+Nb (wt.%: 0.023C, 0.161N, 1.34Mn, 0.48Si, 18.1Cr, 12.5Ni, 2.82Mo, 0.11Nb).

## 5. Conclusions

Precipitation of the modified Z-phase ((Nb,V)CrN) in the AISI 316LN+Nb+V steel was experimentally observed in the temperature interval of 750–1050 °C. The kinetics of precipitation of this minor phase was fast. Particles of the modified Z-phase were already detected after annealing for 1 h at annealing temperatures of 850 and 950 °C. Particles nucleated on both MX ((Nb,V)N) particles and from solid solution.

Results of diffraction studies on the particles of the modified Z-phase confirmed the existence of the tetragonal unit cell already after 1 h annealing. However, streaking along [001]* direction in the diffraction pattern suggested that there were defects along the c axis of the elementary unit cell. These defects were probably related to limited mobility of solute atoms during the formation of the double layer structure of the modified Z-phase. Prolongation of annealing time led to the disappearance of these diffraction effects.

The experimentally determined solvus temperature of the modified Z-phase in the AISI 316LN+Nb+V steel at 1050 °C was in good agreement with the Thermocalc calculations. It was approximately 200 °C lower than that of the Z-phase (NbCrN) in niobium- and nitrogen-bearing austenitic steels.

Residual content of vanadium (0.03 wt.%) in the HR3C steel led to the precipitation of the modified Z-phase containing approximately 3 wt.% of vanadium. Thermocalc calculations suggested the solvus temperature of this phase to be approximately 1200 °C. Particles preferentially nucleated on MX ((Nb,V)N) particles.

These results prove that dissolution of vanadium in the modified Z-phase in austenitic steels causes lowering of its solvus temperature. The decrease in the solvus temperature is dependent on the vanadium content in austenitic steels.

**Author Contributions:** V.V.: TEM investigations, conceptualization, writing—review and editing; J.H. (Jan Holešinský): TEM investigations, TTP diagram construction, thermodynamic calculations, writing—original draft; Z.K.: data curation, validation; R.P.: visualisation, TEM investigations, data curation; K.H.: thermodynamic calculations, data curation; P.V.: funding acquisition, writing—review and editing; J.H. (Josef Hlinka): thermodynamic calculations, data curation. All authors have read and agreed to the published version of the manuscript.

**Funding:** This research was funded by the TACR project no. TK03020055, Research on creep behaviour and verification of brittle-fracture properties of austenitic steels for thermal power plant blocks with USC steam parameters.

**Data Availability Statement:** Not applicable.

**Acknowledgments:** This paper was an output of the project "TK03020055 Research on creep behaviour and verification of brittle-fracture properties of austenitic steels for thermal power plant blocks with USC steam parameters" and the student grant competition project "SP2023/049 The influence of production parameters and operating conditions on the microstructure and utility properties of metallic materials". R.P. is grateful for financial support from the project no. 00424/2022/RRC funded by the Moravian–Silesian Region "Support for gifted students of doctoral studies at VŠB-TUO".

**Conflicts of Interest:** The authors declare no conflict of interest.

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
