# Peer review of "The Effect of Vanadium on Modified Z-Phase Characteristics in Austenitic Steels"

_crystals, doi:10.3390/cryst13040676_

Round 1
Reviewer 1 Report
The submitted manuscript discusses the effect of V on modified Z-phase characteristics in Austenitic steels. This study can contribute to the science by improving the high temperature applications of the investigated alloy. The manuscripts is well written and the English language is fine and minor corrections are needed. The introduction covers the topic adequately and the methods are well described. However, the results need to be logically presented.
The results section must start with the microstructural analyses of the investigated alloys and not by simulating the spot diffraction patterns.
The size of the rod in Linee 122 is unknown. Authors must include the dimensions of the forged rods.
Line 162 contains a website. The reviewer wonders why there is a link in the text. Furthermore, the link is not working and says the page is not found!!!!!
The resolution of Figure 2 is very poor. Authors are encouraged to imrpve the resolution of this figure and other if necessary.
The focus in Figure 4 (a) is poor and need to be improved.
Improve the reolution of Figures 5 and 6.
The rupture exposure period (55500h) seems to be not logical since it equals to around 6 years. The same applies to the 58950h exposure time. Authors must revise the related information.
The thermodynamic simulation of the equilibrium diagrams is not well discussed.
Reviewer 2 Report
The paper is well-written, and the science behind the research is solid. I really like the whole explanation and presentation of identifying the Z-phase with diffraction patterns, the authors did a good job describing the problem. There are just some minor issues that need to be addressed, otherwise, I think this is a very good paper.
The sample from figure 11 isn't well explained and is not mentioned in the experimental part. Also, the term primary Z-phase would indicate precipitation from the melt, not from austenite.
The graph in figure 7 is in seconds, maybe hours would be better.
Some markings of phases in figure 9 would further improve the presentation, in my opinion, but it is not necessary.
I was a bit confused in the conclusion part line 346, the commercial cast of the HR3C, was it a casting? Not a wrought product?
Otherwise I think the paper will be a grat addition to Crystals.
Reviewer 3 Report
The paper is generally well written and can be considered for publication. However, it looks rather than technical report than scientific paper. As a Reviewer, I would like to express my positive opinion. Below, are minor comments:
a/ Fig. 2 is fuzzy - please improve quality
b/ please standardize all diagrams
c/ There is a lack of linking to influence of obtained modified Z-phase on the mechanical properties. Can Authors discuss this issue?
d/ Above question is also one doubt referred to the title. Here is written - Effect....but effect on? please consider modification and / or add more discussion in paper about effects
Round 2
Reviewer 1 Report
The revised manuscript is fine and can be accepted in the current form